# The role of lactate dehydrogenase in hospitalized patients, comparing those with pulmonary versus non-pulmonary infections: A nationwide study

Amit Frenkel[1]*, Adi Shiloh[2], Beatrice Azulay[3], Victor Novack[2,4], Moti Klein[1], Jacob Dreiher[5]

1 General Intensive Care Department, Soroka University Medical Center, and The Faculty of Health Sciences, Ben-Gurion University of the Negev, Beer-Sheva, Israel, 2 Clinical Research Center, Soroka University Medical Center, and The Faculty of Health Sciences, Ben-Gurion University of the Negev, Beer-Sheva, Israel, 3 Division of Anesthesiology and Critical Care, Soroka University Medical Center and the Faculty of Health Sciences, Ben-Gurion University of the Negev, Beer-Sheva, Israel, 4 Anesthesia, Critical Care and Pain Medicine, Beth Israel Deaconess Medical Center, Harvard Medical School, Boston, MA, United States of America, 5 Hospital Administration, Soroka University Medical Center, and The Faculty of Health Sciences, Ben-Gurion University of the Negev, Beer-Sheva, Israel

* frenkela@clalit.org.il

## Abstract

### Background

Lactic dehydrogenase reflects target organ damage, and is associated with mortality in patients with infectious diseases.

### Objective

The purpose of this study was to examine associations of serum lactic dehydrogenase levels with mortality, target organ damage and length of hospital stay in adults with pulmonary and non-pulmonary infections.

### Methods

This nationwide retrospective cohort study comprised patients admitted with infections, to medical and surgical departments in eight tertiary hospitals during 2001–2020. Patients with available serum lactic dehydrogenase levels on admission and one week after were included, and stratified by the source of their infection: pulmonary vs. non-pulmonary. Associations of lactic dehydrogenase levels with mortality and target organ damage were analyzed using multivariable logistic regression models. Quantile regression was used for multivariable analysis of the median length of stay.

### Results

The study included 103,050 patients (45.4% male, median age: 69 years); 44,491 (43.1%) had pulmonary infections. The median serum lactic dehydrogenase levels on admission

**Data Availability Statement:** Data cannot be shared publicly because of Ethics committee

regulations. Data are available from the Institutional Ethics Committee (Soroka university medical center, contact via Mrs. Naomi Amichai, email: NaomiAm@clalit.org.il).

**Funding:** The author(s) received no specific funding for this work.

**Competing interests:** The authors have declared that no competing interests exist.

were higher in patients with pulmonary than non-pulmonary infections (418 vs. 385 units per liter (U/L), p<0.001). In a multivariable logistic regression model, elevated serum lactic dehydrogenase levels (480–700 U/L, 700–900 U/L and >900 U/L), compared with <480 U/L, were associated with in-hospital mortality (OR = 1.81, 2.85 and 3.69, respectively) and target organ damage (OR = 1.19, 1.51 and 1.80, respectively). The median stay increased with increasing elevated lactic dehydrogenase levels (+0.3, +0.5 and +0.4 days, respectively). Among patients with lactic dehydrogenase levels >900 U/L, mortality, but none of the other examined outcomes, was greater among those with pulmonary than non-pulmonary infections.

## Conclusions

Among hospitalized patients with infectious diseases, lactic dehydrogenase levels were associated with mortality and target organ damage, and were similar in patients with pulmonary and non-pulmonary infections. Among patients with lactic dehydrogenase levels >900 U/L, mortality was prominently higher among those with pulmonary than non-pulmonary infections.

## Introduction

Lactate dehydrogenase (LDH) is a cytoplasmic enzyme that catalyzes the conversion of lactate to pyruvate, an important step in energy production in cells. LDH exhibits five isomeric forms (LDH-1, 2, 3, 4, and 5); each has a slightly different structure and is found at particular concentrations in the various tissues [1]. LDH serum levels are elevated in various diseases, in proportion to the extent of cellular damage. Malignancies, and neuromuscular, rheumatologic, hematologic, endocrine, renal and infectious diseases may all be characterized by a significant increase in LDH. The classical well-known involvement of serum LDH in infectious diseases was described many years ago. Zaman et al. reported that serum LDH levels can be a useful marker of Pneumocystis carinii pneumonia in patients infected with the human immunodeficiency virus [2]. Garba et al. reported that serum LDH activity is a potentially valuable enzymatic marker of acute, uncomplicated Plasmodium falciparum malaria infection [3]. Sharma et al. reported on the role of serum LDH in the diagnosis of Mycobacterium tuberculosis [4].

Despite the above, as many types of body cells (of the heart, kidneys, liver, lungs and muscles) contain LDH, serum LDH levels may be increased in a wide range of non-pulmonary as well as pulmonary infections. Ewig et al were among the first to describe, in 1995, that above normal LDH serum values were closely associated with mortality, in patients with community-acquired pneumonia [5]. In 2018, Jun et al. reported an association of higher levels of serum LDH with higher 28-day mortality, in patients with infection of any source [6]. Similarly, Zein J et al. concluded that LDH is a marker of cell injury, and that failed improvement in LDH levels at 48 hours can predict mortality in patients with sepsis [7].

The main aim of the current study was to assess associations of serum LDH levels with mortality, target organ damage and length of stay; and to compare the findings between patients with pulmonary and non-pulmonary infections. Serum LDH levels were found to be associated with the outcomes examined. At highly elevated LDH levels, mortality was found to differ between those with pulmonary and non-pulmonary infections.

## Materials and methods

### Study population

We conducted a multicenter population-based retrospective cohort study of patients hospitalized at one of eight academic medical centers owned by Clalit Health Services (CHS). CHS is the largest healthcare provider organization in Israel, with over 4.5 million members. All CHS members hospitalized between December 2001 and October 2020 with a primary diagnosis of infectious disease based on the International Classification of Diseases, 9th revision clinical modification (ICD-9-CM) diagnoses codes were included. To avoid including patients with nosocomial infections, which may possess unique and distinguished characteristics, only patients with infections documented within the first day of hospital admission were included. Hospitalization length of stay was restricted to 3–30 days. This limited the cohort to patients with a substantial illness that required hospitalization, whether acute or sub-acute.

According to the methodology developed by Martin et al. [8], sources of infection were categorized according to the ICD-9-CM codes: pulmonary, cardiovascular, skin and soft tissue, urinary, nervous system, gastrointestinal and peritoneum, and bone and joints. Patients with bacteremia or other infections whose source could not be determined were excluded from the analysis.

### Data extraction and sources

The study was conducted using the data warehouse of CHS, as in similar investigations [9, 10]. The following data were collected: demographics (age, sex), hospitalization details (length of stay, intensive care unit (ICU) transfers), underlying medical conditions according to the chronic disease registries of CHS, chronic medications purchased three months prior to hospitalization based on the Anatomical Therapeutic Chemical (ATC) coding of the World Health Organization's classification system, and laboratory blood results from the index hospitalization. Target organ damage was defined using the methodology described by Martin et al. [8], based on ICD-9-CM codes for acute organ system failure (respiratory: 518.5–518.53, 518.81, 518.82, 518.84, 518.85, 786.09, 799.1, Z79.7, Z96.70–Z96.72; cardiovascular: 427.5, 458.0, 458.8, 458.9, 785.5, 785.50–785.52, 785.59, 785.591, 796.3; renal: 580, 580.0, 580.4, 580.81, 580.89, 580.9, 584, 584.5–584.9, 585, Z39.95; liver: 570, 572.2, 573.3; hematologic: 286.6, 286.9, 287.30–287.32, 287.4, 287.49, 287.5; metabolic: 276.2; neurologic: 293, 293.0, 293.1, 293.9, 348.1, 348.3, 348.31, 348.39, 780.01, 780.09, Z89.14; nonspecific: 995.92).

The main exposure variables were serum levels of LDH from the time of diagnosis to one week later. The first LDH result was used to examine associations with the outcome variables. LDH levels were categorized as <480 U/L, 480–700 U/L, 701–900 U/L and >900 U/L.

Outcome variables included 30-day mortality, target organ damage during hospitalization (one or more), and median hospitalization length of stay (accounting only for patients who survived the hospitalization).

Data were extracted from the CHS database using a data-sharing platform powered by MDClone [11]. The study was conducted according to the guidelines of the Declaration of Helsinki, and approved by the Institutional Review Board of Soroka Medical Center (protocol number 0108-16-SOR). No informed consent was needed, and all data were fully anonymized before we accessed them.

### Statistical analysis

When appropriate, univariate comparisons were made using Chi-square test or Fisher's exact test for categorical variables, and using the independent sample T-test or Mann-Whitney test

for quantitative or ordinal variables. Multivariable logistic regression was used to model the factors associated with 30-day mortality and target organ damage, and quantile regression was used for assessing factors associated with the median hospitalization length of stay. Variables were evaluated as potential confounders according to the results of the univariate analysis (p<0.1) or their clinical importance. For the point estimates of odds ratios or coefficients in the quantile regression, 95% confidence intervals were calculated, and rounded outwards. Due to the large sample size, a two-sided p-value of < = 0.01 was considered statistically significant. IBM SPSS software, version 26.0, was used for the statistical analysis.

## Results

The study included 103,050 patients (45.4% male, median age: 69 years). Of these, 44,491 (43.1%) had pulmonary infections. Compared to patients with non-pulmonary infections, those with pulmonary infections were slightly older, more likely to be male, and more likely to be diagnosed with certain comorbidities: congestive heart failure, chronic obstructive pulmonary disease, asthma and atrial fibrillation (Table 1). Among patients with pulmonary compared to non-pulmonary infections, in-hospital mortality was higher (6.9% vs. 3.4%, p<0.001); and frequencies were higher of transfers to the ICU (2.1% vs. 1.1%, p<0.001) and of target organ damage (14.0% vs 11.5%, p<0.001). The most common target organ damage was renal injury, in both groups. Among patients with pulmonary versus non-pulmonary infections, the median serum LDH level on admission was higher (418 U/L vs. 385 U/L, p<0.001). The proportions of patients with elevated LDH were higher among those with pulmonary than non-pulmonary infections (25.7 vs. 19.6%, 5.1 vs. 3.8% and 2.6 vs. 2.2%, for LDH levels of 480–700 U/L, 701–900 U/L and 900 U/L, respectively, p<0.001 for all).

In multivariable logistic regression models (Table 2 and Fig 1), the factors that were associated with 30-day mortality, both in patients with pulmonary infections and in patients with non-pulmonary infections, were age, target organ damage, laboratory abnormalities and underlying comorbidities. In those models, elevated serum LDH was associated with in-hospital mortality (OR = 1.81, 2.85 and 3.69 for LDH levels of 480–700 U/L, 701–900 U/L and >900 U/L, respectively, compared to LDH levels <480 U/L). The results were similar for patients with pulmonary infections and non-pulmonary infections, although the ORs were somewhat higher for patents with pulmonary infection (Fig 1). No significant interaction between LDH levels and the source of infection was noted (p = 0158, p = 0.811 and p = 0.078 for the all three categories of LDH levels).

Target organ damage was associated with age, male sex, certain comorbidities and laboratory abnormalities (Table 3). The multivariable logistic regression model showed a higher probability of target organ damage among patients with elevated LDH levels (OR = 1.19, 1.51 and 1.80, for LDH levels of 480–700 U/L, 701–900 U/L and >900 U/L, respectively, compared to LDH levels <480 U/L). Again, the results were similar for patients with pulmonary and non-pulmonary infections, and significant interactions between LDH levels and the source of infection were not found (p>0.482, p = 0.834 and p = 0.991 for the three elevated categories of LDH levels).

A multivariable model for the median length of stay, using quantile regression, is presented in Table 4. The length of stay was associated with age, male sex, laboratory abnormalities and underlying comorbidities. Elevated serum LDH was associated with a slightly longer median hospital stay (+0.3 days, +0.5 days and +0.4 days, for LDH levels of 480–700 U/L, 701–900 U/L and >900 U/L, respectively). The increase was similar for pulmonary and non-pulmonary infections. The results were similar in a sensitivity analysis of patients with the 75th percentile of length of stay.

**Table 1. Demographic characteristics and hospitalization details.**

|  | Infection from pulmonary source | Infection from other sources | P-value |
|---|---|---|---|
|  | n = 44,491 | n = 50,486 |  |
| Age, years, mean ± SD | 70.7 ± 17.3 | 68 ± 19.4 | <0.001 |
| Male sex | 23226 (52.2) | 23568 (46.7) | <0.001 |
| Diabetes mellitus | 16147 (36.3) | 18821 (37.3) | 0.002 |
| Ischemic heart disease | 16070 (36.1) | 15516 (30.7) | <0.001 |
| History of stroke | 2348 (5.3) | 2954 (5.9) | <0.001 |
| Hematologic malignancy | 2350 (5.3) | 1812 (3.6) | <0.001 |
| Solid malignancy | 24835 (55.8) | 27775 (55.0) | 0.010 |
| Liver cirrhosis | 534 (1.2) | 919 (1.8) | <0.001 |
| Congestive heart failure | 10899 (24.5) | 8434 (16.7) | <0.001 |
| Chronic renal failure | 10151 (22.8) | 11435 (22.6) | 0.540 |
| COPD | 15304 (34.4) | 6200 (12.3) | <0.001 |
| Hypertension | 28915 (65) | 31893 (63.2) | <0.001 |
| Asthma | 8111 (18.2) | 4649 (9.2) | <0.001 |
| Atrial fibrillation/flutter | 9596 (21.6) | 8114 (16.1) | <0.001 |
| Overall mortality | 26651 (59.9) | 26031 (51.6) | <0.001 |
| In-hospital | 1846 (6.9) | 890 (3.4) | <0.001 |
| 30-day mortality | 1782 (6.7) | 1779 (6.8) | 0.500 |
| ICU transfer | 934 (2.1) | 547 (1.1) | <0.001 |
| Hospitalization length of stay, median (IQR) | 5 (3–7) | 5 (3–7) | <0.001 |
| Any target organ damage | 6230 (14) | 5810 (11.5) | <0.001 |
| Respiratory | 2341 (5.3) | 522 (1) | <0.001 |
| Vascular | 303 (0.7) | 313 (0.6) | 0.240 |
| Renal | 3074 (6.9) | 4194 (8.3) | <0.001 |
| Liver | 59 (0.1) | 336 (0.7) | <0.001 |
| Hematologic | 617 (1.4) | 631 (1.2) | 0.06 |
| Metabolic | 358 (0.8) | 71 (0.1) | <0.001 |
| CNS | 24 (0.1) | 79 (0.2) | <0.001 |
| Hb < 7g/L | 495 (1.1) | 569 (1.1) | 0.9 |
| WBC > 15,000 mm3 | 13476 (30.6) | 17065 (34.0) | <0.001 |
| WBC < 5,000 mm3 | 6019 (13.9) | 6670 (13.4) | 0.06 |
| PLT < 50,000 / mm3 | 742 (1.7) | 622 (1.3) | <0.001 |
| Glucose > 200 mg/dl | 10175 (23.3) | 10186 (20.6) | <0.001 |
| Glucose < 50 mg/dl | 545 (1.3) | 719 (1.5) | 0.008 |
| Creatinine (max), mg/dL, median (IQR) | 1 (0.8–1.4) | 1 (0.8–1.4) | <0.001 |
| Maximal Cr/first Cr > 2 | 515 (1.2) | 451 (0.9) | <0.001 |
| Albumin < 2.5 g/l | 3437 (8.0) | 4402 (9.2) | <0.001 |
| Median LDH on admission (U/L, IQR) | 418 (344–522) | 385 (316–483) | <0.001 |
| Median maximal LDH (U/L, IQR) | 442 (360–560) | 409 (332–518) | <0.001 |

The data are presented as number (%), unless stated otherwise.

COPD, chronic obstructive pulmonary disease

ICU, intensive care unit

IQR, interquartile range

CNS, central nervous system

Hb, hemoglobin

WBC, white blood cells

PLT, platelets

Cr, creatinine

LDH, Lactic dehydrogenase

**Table 2. Factors associated with 30-day mortality (multivariable logistic regression).**

| | All patients | Infection from pulmonary source | Infection from non-pulmonary source |
|---|---|---|---|
| | N = 94,977 | N = 44,491 | N = 50,486 |
| | Odds Ratio | Odds Ratio | Odds Ratio |
| | (95% CI) | (95% CI) | (95% CI) |
| LDH (categories) U/L | | | |
| < 480 | | Reference category | |
| 480–700 | 1.81 (1.63–2.02) | 1.64 (1.49–1.80| | 1.82 (1.63–2.03) |
| 701–900 | 2.85 (2.37–3.42) | 2.74 (2.34–3.21) | 2.90 (2.41–3.48) |
| >900 | 3.69 (2.97–4.58) | 4.77 (3.93–5.78) | 3.73 (3.01–4.63) |
| Age (per year) | 1.07 (1.07–1.07) | 1.07 (1.07–1.08) | 1.07 (1.06–1.07) |
| Target organ damage | 1.62 (1.5–1.76) | 1.69 (1.53–1.88) | 1.53 (1.35–1.72) |
| Glucose < 50 mg/dl | 1.43 (1.34–1.54) | 1.41 (1.28–1.55) | 1.47 (1.32–1.64) |
| Glucose > 200 mg/dl | 1.50 (1.22–1.83) | 1.89 (1.42–2.53) | 1.23 (0.92–1.64) |
| WBC > 150,000 /mm3 | 1.44 (1.34–1.54) | 1.62 (1.48–1.77) | 1.23 (1.11–1.36) |
| WBC < 5000 / mm3 | 0.72 (0.65–0.81) | 0.77 (0.67–0.88) | 0.67 (0.57–0.81) |
| Hb < 7 g/l | 1.79 (1.43–2.24) | 1.61 (1.18–2.21) | 1.99 (1.45–2.72) |
| PLT < 50,000 / mm3 | 2.11 (1.66–2.69) | 2.33 (1.67–3.25) | 1.98 (1.39–2.81) |
| Albumin < 2.5 g/l | 6.08 (5.64–6.56) | 6.21 (5.58–6.91) | 5.95 (5.34–6.62) |
| Maximal Cr/first Cr > 2 | 4.41 (3.69–5.28) | 5.68 (4.48–7.20) | 5.95 (5.34–6.62) |
| Ischemic heart disease | 0.94 (0.87–1.00) | 0.89 (0.81–0.98) | 1.00 (0.90–1.11) |
| Previous stroke | 1.52 (1.37–1.70) | 1.60 (1.38–1.86) | 1.41 (1.21–1.66) |
| Cancer | 1.19 (1.11–1.27) | 1.20 (1.10–1.30) | 1.17 (1.06–1.29) |
| Congestive heart failure | 1.36 (1.26–1.47) | 1.31 (1.18–1.44) | 1.45 (1.29–1.63) |
| Hypertension | 0.82 (0.76–0.89) | 0.76 (0.69–0.85) | 0.90 (0.80–1.02) |
| Asthma | 0.80 (0.72–0.89) | 0.77 (0.68–0.87) | 0.87 (0.73–1.04) |
| Atrial fibrillation | 1.14 (1.06–1.23) | 1.03 (0.93–1.13) | 1.32 (1.18–1.48) |
| Pulmonary infection | 1.72 (1.58–1.88) | NA | NA |

LDH, Lactic dehydrogenase

WBC, white blood cells

Hb, hemoglobin

PLT, platelets

Cr, creatinine

NA, not applicable

## Discussion

The main finding of our study is that elevated LDH levels in hospitalized patients with infectious diseases were associated with a higher mortality rate and more target organ damage, regardless of whether the source of infection was pulmonary or non-pulmonary. Additionally, our univariate non-adjusted analysis revealed a higher median LDH level in patients with pulmonary infections (418 U/L) compared to those with non-pulmonary infections (385 U/L), with a p-value of <0.001.

Efforts have been invested to identify biomarkers that could predict mortality in patients hospitalized with infectious diseases. Some of the "classic" biomarkers are white blood cell (WBC) count, and albumin and LDH levels. Newer biomarkers include C-reactive protein (CRP) and procalcitonin. In some circumstances, the classic biomarkers have been shown to predict mortality with similar accuracy as the newer biomarkers. For example, in 2020, Tascini

(a)

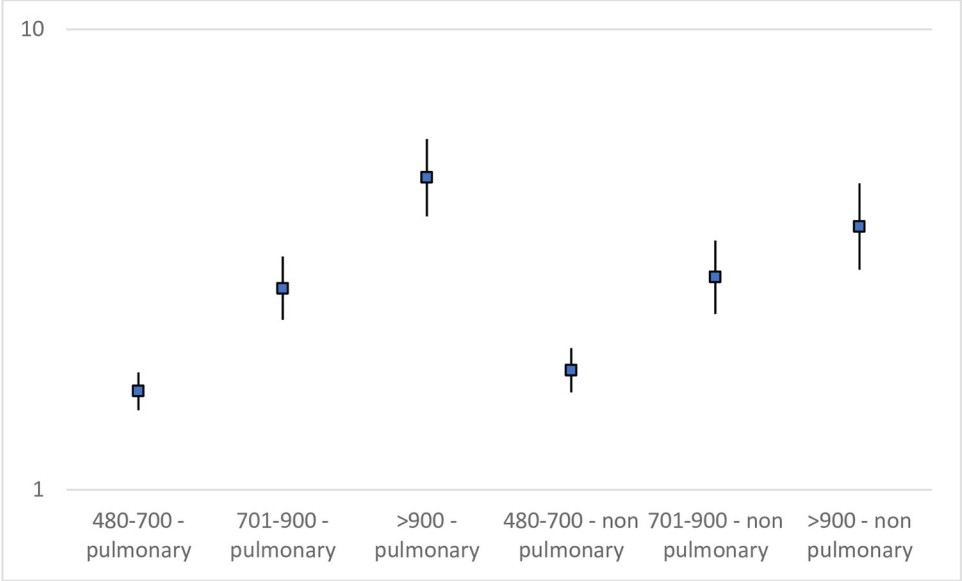

(b)

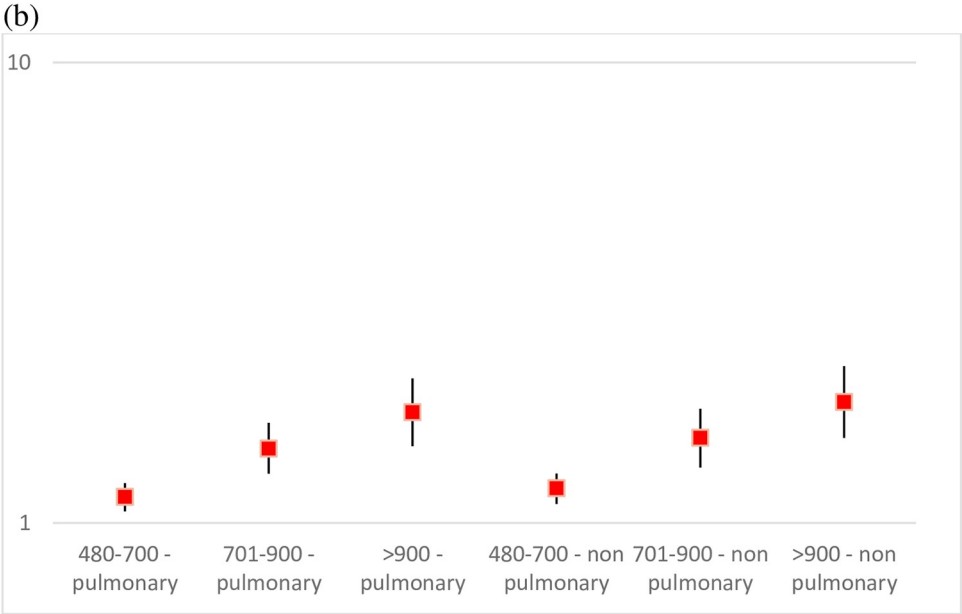

**Fig 1.** Odds ratios for (a) mortality and (b) target organ damage, by source of infection and LDH levels. The y axis is in a logarithmic scale.

et al. [12] showed that WBC, as well as procalcitonin and CRP on admission, was a good predictor for mortality in patients with Staphylococcus aureus infective endocarditis. In 2021, Yahua et al. [13] showed that the CRP-to-albumin ratio predicted mortality in patients with sepsis due to burns, similarly as albumin levels themselves predict mortality. LDH is a widely available biomarker. However, other than the two papers mentioned above, published in 1995 and 2018 [13, 14], no other recently published research investigated its value in predicting mortality in hospitalized patients with infectious diseases. Furthermore, to the best of our

**Table 3. Factors associated with target-organ damage (multivariable logistic regression).**

| | All patients | Infection from pulmonary source | Infection from non-pulmonary source |
|---|---|---|---|
| | N = 94,977 | N = 44,491 | N = 50,486 |
| | Odds Ratio | Odds Ratio | Odds Ratio |
| | (95% CI) | (95% CI) | (95% CI) |
| LDH (categories) U/L | | | |
| < 480 | | Reference category | |
| 480–700 | **1.19 (1.10–1.28)** | **1.14 (1.06–1.22** | **1.19 (1.10–1.28)** |
| 701–900 | **1.51 (1.31–1.75)** | **1.45 (1.28–1.65)** | **1.53 (1.32–1.77)** |
| >900 | **1.80 (1.50–2.15)** | **1.74 (1.47–2.06)** | **1.83 (1.53–2.19)** |
| Age (per year) | **1.01 (1.01–1.01)** | **1.00 (1.06–1.07)** | **1.01 (1.01–1.01)** |
| Male sex | **1.11 (1.06–1.16)** | **1.10 (1.03–1.17)** | **1.13 (1.06–1.21)** |
| Glucose < 50 mg/dl | **1.33 (1.27–1.40)** | **1.34 (1.25–1.43)** | **1.33 (1.23–1.43)** |
| Glucose > 200 mg/dl | **1.53 (1.31–1.80)** | **1.51 (1.19–1.91)** | **1.55 (1.25–1.92)** |
| WBC > 150,000 / mm$^3$ | **1.22 (1.16–1.28)** | **1.32 (1.23–1.41)** | **1.12 (1.05–1.20)** |
| WBC < 5,000 / mm$^3$ | **1.28 (1.21–1.37)** | **1.29 (1.18–1.41)** | **1.28 (1.17–1.40)** |
| Hb < 7 g/l | **1.24 (1.02–1.50)** | 1.09 (0.83–1.44) | **1.40 (1.08–1.82)** |
| PLT < 50,000 / mm$^3$ | **3.88 (3.27–4.61)** | **3.87 (3.02–4.96)** | **3.93 (3.10–4.99)** |
| Albumin < 2.5 g/l | **1.79 (1.67–1.92)** | **1.90 (1.72–2.11)** | **1.67 (1.52–1.84)** |
| Maximal Cr/first Cr > 2 | **3.70 (3.17–4.32)** | **3.68 (2.99–4.52)** | **3.75 (2.98–4.73)** |
| Cancer | **1.09 (1.04–1.14)** | **1.12 (1.05–1.20)** | 1.05 (0.99–1.13) |
| Liver cirrhosis | **1.63 (1.41–1.90)** | **1.49 (1.18–1.89)** | **1.74 (1.43–2.11)** |
| Congestive heart failure | **1.19 (1.12–1.25)** | **1.15 (1.07–1.24)** | **1.26 (1.16–1.36)** |
| Chronic renal failure | **2.62 (2.49–2.75)** | **2.30 (2.14–2.47)** | **2.99 (2.79–3.21)** |
| COPD | **1.15 (1.10–1.22)** | **1.18 (1.11–1.26)** | 1.09 (0.99–1.19) |
| Hypertension | **1.22 (1.15–1.30)** | **1.18 (1.09–1.28)** | **1.27 (1.17–1.39)** |
| Atrial fibrillation | **1.06 (1.00–1.12)** | **1.11 (1.03–1.19)** | 1.02 (0.93–1.10) |
| Pulmonary infection | **1.72 (1.58–1.88)** | NA | **NA** |

LDH, lactic dehydrogenase

WBC, white blood cells

Hb, hemoglobin

PLT, platelets

Cr, creatinine

COPD, chronic obstructive pulmonary disease

NA, not applicable

knowledge, no published study has compared the mortality predictive value of LDH in pulmonary and non-pulmonary infections.

We analyzed separately the mortality predictive value of LDH in patients with pulmonary and non-pulmonary infections. The rationale of such analysis is that though LDH presents in almost all body tissues, it presents in much larger quantities in the lungs than in other organs. A study based on a nationally representative United States database found that lung infections were the most commonly listed (34%) source of infection in hospitalized patients, followed by kidney, urinary tract, and bladder infections; cellulitis; and abdominal and rectal infections [14]. Notably, except for the kidneys, these non-pulmonary organs do not contain substantial concentrations of LDH. Thus, stratifying our sample by pulmonary and non-pulmonary infections improved the accuracy of our analysis, and accounted for the higher values of LDH that are expected in infections arising from pulmonary origin, regardless of infection severity. In

**Table 4. Factors associated with median length of stay (multivariable quantile regression).**

| | All patients | Infection from pulmonary source | Infection from non-pulmonary source |
|---|---|---|---|
| | N = 94,977 | N = 44,491 | N = 50,486 |
| | B (95% CI) | B (95% CI) | B (95% CI) |
| LDH (categories) U/L | | | |
| < 480 | Reference category | | |
| 480–700 | **0.27 (0.20–0.34)** | **0.15 (0.08–0.21)** | **0.26 (0.18–0.33)** |
| 701–900 | **0.54 (0.39–0.70)** | **0.55 (0.41–0.68)** | **0.58 (0.41–0.74)** |
| >900 | **0.44 (0.25–0.64)** | **0.60 (0.41–0.79)** | **0.43 (0.22–0.64)** |
| Age (per year) | 0.01 (0.01–0.02) | **0.01 (0.01–0.01)** | **0.01 (0.01–0.02)** |
| Male sex | -0.05 (-0.09- -0.01) | **-0.16 (-0.22- -0.11)** | **0.07 (0.01–0.13)** |
| Any target organ damage | **0.71 (0.64–0.77)** | **0.87 (0.78–0.96)** | **0.56 (0.46–0.66)** |
| Glucose < 50 g/dl | **0.61 (0.56–0.66)** | **0.58 (0.51–0.65)** | **0.64 (0.56–0.72)** |
| Glucose > 200 g/dl | **1.57 (1.37–1.76)** | **1.75 (1.47–2.04)** | **1.46 (1.21–1.75)** |
| WBC > 150,000 / mm$^3$ | **0.54 (0.49–0.59)** | **0.51 (0.45–0.57)** | **0.56 (0.50–0.63)** |
| WBC < 5,000 / mm$^3$ | **0.55 (0.49–0.61)** | **0.51 (0.43–0.59)** | **0.58 (0.49–0.66)** |
| Hb < 7 g/l | **1.52 (1.28–1.75)** | **1.54 (1.21–1.87)** | **1.51 (1.18–1.84)** |
| PLT < 50,000 / mm$^3$ | 0.04 (-0.20–0.29) | -0.04 (-0.38–0.30) | 0.12 (-0.24–0.47) |
| Albumin < 2.5 g/l | **3.26 (3.17–3.34)** | **3.17 (3.05–3.30)** | **3.36 (3.24–3.47)** |
| Maximal Cr/first Cr > 2 | **3.97 (3.72–4.22)** | **3.89 (3.54–4.24)** | **4.22 (3.86–4.58)** |
| Liver cirrhosis | **0.10 (0.01–0.19)** | **-0.29 (-0.56- -0.03)** | **-0.34 (-0.58- -0.10)** |
| Congestive heart failure | -0.33 (-0.51–0.16) | **0.36 (0.29–0.43)** | **0.31 (0.22–0.39)** |
| Atrial fibrillation | 0.34 (0.28–0.40) | **0.25 (0.18–0.32)** | **0.16 (0.07–0.25)** |
| Pulmonary infection | -0.05 (-0.11–0.01) | **NA** | **NA** |

LDH, lactic dehydrogenase

WBC, white blood cells

Hb, hemoglobin; PLT, platelets

Cr, creatinine

target organs other than the lungs, LDH values are lower, and thus may more closely reflect the severity of the infection itself. Our finding of higher median LDH values among patients whose source of infection was pulmonary compared to non-pulmonary supports this notion.

Theoretically, due to the higher values of LDH expected in pulmonary infections, we could presume that the mortality OR for each LDH range would be lower in the pulmonary infection group. This is because the LDH values do not reflect only the severity of the infection, as in non-pulmonary infections, but also secretion of larger amounts of LDH upon injury. However, mortality OR rates were similar between patients with pulmonary and non-pulmonary infection, for two of the three elevated LDH ranges examined. Among patients with the highest LDH values (> 900 U/L), the OR for mortality was considerably higher among those with pulmonary than non-pulmonary infections (4.77 vs. 3.73). Possibly, particularly high LDH values among patients with pulmonary infection indicate additional severe lung parenchymal damage, such as increased incidence of acute respiratory distress syndrome. This may explain the closer association between higher mortality and high LDH values, in patients with pulmonary compared to non-pulmonary infections.

Although leukopenia is widely recognized in the literature as a marker for mortality in patients with infection [15], our study yielded unexpected results, as leukopenia was not found to be associated with mortality. Instead, it was linked to increased target organ damage. In contrast, anemia and thrombocytopenia, which have been consistently associated with adverse

outcomes in previous studies [16], were found to be correlated with both mortality and target organ damage. Similarly, low levels of albumin were significantly associated with all outcomes, as previously demonstrated in a recently published research [10].

It is worth noting that the current study did not involve patients with COVID-19 infection, despite data being collected up until 2020. However, it is noteworthy that recent research has highlighted a significant association between high LDH values and disease severity and mortality among COVID-19 patients [17].

Our study has a number of limitations. First, only patients with an infectious disease for whom LDH levels were taken were included in the analysis. As the hospitals from which the data were extracted do not have orderly protocols for testing LDH, as part of the hospitalization process, testing may have only been performed for the more severe patients. Therefore, the results do not accurately reflect the entire population of patients hospitalized with infections. In this context, it is worth noting that we took the first LDH measurement of each patient. Thus, LDH values were not taken at the same point in time relative to the onset of disease. Second, we categorized various diseases as "pulmonary infections", and did not perform a sub-analysis of these diseases. Nor did we perform a sub-analysis of patients whose source of infection was renal. This is despite the relatively high frequency of renal infections in hospitalized patients, and the expectation that renal damage results in relatively high LDH values, similarly to pulmonary infections. A key strength of the present study is the size of the population and its distribution: our cohort contains data collected over about 20 years, from eight hospitals throughout Israel.

To the best of our knowledge, this is the largest published study of an analysis that included only patients who were hospitalized due to infectious disease, and for which LDH tests were part of routine blood work during hospitalization.

## Conclusions

We suggest that elevated LDH levels in hospitalized patients with infectious diseases are associated with higher mortality rate and more target organ damage, in both pulmonary and non-pulmonary infections. Additionally, the analysis revealed that patients with pulmonary infections and LDH levels greater than 900 U/L had higher mortality odds within a 30-day timeframe. These findings support the utility of LDH testing as a valuable bedside tool for estimating mortality odds in patients with infectious diseases.

## Supporting information

**S1 File.**
(DOCX)

## Author Contributions

**Conceptualization:** Amit Frenkel.

**Data curation:** Beatrice Azulay.

**Formal analysis:** Adi Shiloh.

**Methodology:** Victor Novack.

**Resources:** Jacob Dreiher.

**Supervision:** Moti Klein, Jacob Dreiher.

**Writing – original draft:** Amit Frenkel.

**Writing – review & editing:** Jacob Dreiher.

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
