## [Decision Letter · Decision Letter 0]

20 Jan 2023

PONE-D-22-32402­­The role of lactate dehydrogenase in hospitalized patients, comparing those with pulmonary versus non-pulmonary infections: A nationwide studyPLOS ONE

Dear Dr. Frenkel,

Thank you for submitting your manuscript to PLOS ONE. After careful consideration, we feel that it has merit but does not fully meet PLOS ONE’s publication criteria as it currently stands. Therefore, we invite you to submit a revised version of the manuscript that addresses the points raised during the review process.

We look forward to receiving your revised manuscript.

Kind regards,

Riccardo Nevola, MD, PhD

Academic Editor

PLOS ONE

Journal Requirements:

2. In the ethics statement in the manuscript and in the online submission form, please provide additional information about the patient records/samples used in your retrospective study. Specifically, please ensure that you have discussed whether all data/samples were fully anonymized before you accessed them and/or whether the IRB or ethics committee waived the requirement for informed consent. If patients provided informed written consent to have data/samples from their medical records used in research, please include this information.

Reviewers' comments:

Reviewer's Responses to Questions

**Comments to the Author**

1. Is the manuscript technically sound, and do the data support the conclusions?

Reviewer #1: Yes

Reviewer #2: Partly

Reviewer #3: Yes

2. Has the statistical analysis been performed appropriately and rigorously? 

Reviewer #1: I Don't Know

Reviewer #2: Yes

Reviewer #3: Yes

3. Have the authors made all data underlying the findings in their manuscript fully available?

Reviewer #1: Yes

Reviewer #2: Yes

Reviewer #3: No

4. Is the manuscript presented in an intelligible fashion and written in standard English?

Reviewer #1: Yes

Reviewer #2: Yes

Reviewer #3: Yes

5. Review Comments to the Author

Reviewer #1: a well conducted study with a good number of studied patients and very clearly written paper. I think the practical application will be of limited impact on the management of the hospitalized patients regarding both diagnosis and treatment

Reviewer #2: Thank you for the opportunity to review this interesting study. Authors have performed retrospective review of inpatients admitted with infection between 2001-2020 to understand relationship of lactate dehydrogenase in regarding to outcomes in pulmonary and non-pulmonary infections. I have few suggestions listed below to improve the manuscript:

1. In this study, authors have shown pulmonary and nonpulmonary infections in an unadjusted model (table 1) only. Also, baseline characteristics (table 1) suggests that both the populations are inherently non-comparable without adjusted analysis. In the discussion 'We also found that the median serum LDH level among patients with pulmonary infections was higher than that of patients with non-pulmonary infections (418 U/L vs. 385 U/L, p<0.001).' may not be correctly interpretable. Author should emphasize this limitation.

2. In table 2, please explain, if gender was added in the model ?

3. I would encourage authors to add univariate analysis in the supplementary tables for transparency of selection of variables in the multivariate analysis.

4. In the table 2, 3, and 4, authors have added many biomarkers such as Hb, WBC, PLT, and Albumin. And all these biomarkers have significant associations. However, authors have not discussed the potential role/importance of these markers. I would suggest adding more discussion about these biomarkers.

5. In the conclusion section, 'Additionally, patients with pulmonary infections and LDH levels >900 U/L are at particularly increased risk of death within 30 days. The findings presented here suggest LDH testing as a helpful bedside tool to estimate mortality risk in patients with infectious disease.' I would suggest rephrasing these sentences, as this study did not assess risk but 'increase odds of death'.

6. A number of previous studies and reviews have shown that In patients with COVID-19, the increased LDH level is associated with a higher risk of negative clinical prognosis and higher mortality. Current data included the year 2020 in the analysis. Author should add a caveat specific diseases such as COVID-19.

7. I will suggest to include more references to support the statements made in the discussion. I will also suggest to review the manuscript for grammar and use same fonts for manuscript and all references.

Reviewer #3: Thorough, intelligible review of elevated LD in sepsis with meaningful, interesting results.

Possibly limited clinical utility given the increased LD in different kind of sepsis only modest and would add little to clinical evaluation and management.

There is a considerable literature in this area, which they only summarise but usefully compare to other biomarkers.

**
Note that the Editor reserves further comments at later stages of the manuscript evaluation process.
**

6. PLOS authors have the option to publish the peer review history of their article (what does this mean?). If published, this will include your full peer review and any attached files.

Reviewer #1: No

Reviewer #2: No

Reviewer #3: No

---

## [Author Response · Author response to Decision Letter 0]

23 Feb 2023

Reviewer 1 report: 

We thank the reviewer for the thorough and incisive review of our manuscript. Our responses to the comments are listed below.

Comment # 1

a well conducted study with a good number of studied patients and very clearly written paper. I think the practical application will be of limited impact on the management of the hospitalized patients regarding both diagnosis and treatment

Our response:

We concur with the reviewer's perspective that the article is not intended to alter diagnosis or treatment, but rather to serve as a convenient bedside resource for prognostic purposes.

Reviewer 2 report:

We appreciate the reviewer’s positive evaluation of our manuscript and the important comment, to which we responded below

Thank you for the opportunity to review this interesting study. Authors have performed retrospective review of inpatients admitted with infection between 2001-2020 to understand relationship of lactate dehydrogenase in regarding to outcomes in pulmonary and non-pulmonary infections. I have few suggestions listed below to improve the manuscript:

Comment # 1

In this study, authors have shown pulmonary and nonpulmonary infections in an unadjusted model (table 1) only. Also, baseline characteristics (table 1) suggests that both the populations are inherently non-comparable without adjusted analysis. In the discussion 'We also found that the median serum LDH level among patients with pulmonary infections was higher than that of patients with non-pulmonary infections (418 U/L vs. 385 U/L, p<0.001).' may not be correctly interpretable. Author should emphasize this limitation.

Our response:

We agree with the reviewer that this issue should be emphasized, and edited the discussion section accordingly: 

"The main finding of our study is that elevated LDH levels in hospitalized patients with infectious diseases were associated with a higher mortality rate and more target organ damage, regardless of whether the source of infection was pulmonary or non-pulmonary. Additionally, our univariate non-adjusted analysis revealed a higher median LDH level in patients with pulmonary infections (418 U/L) compared to those with non-pulmonary infections (385 U/L), with a p-value of <0.001."

Comment # 2

In table 2, please explain, if gender was added in the model ?

Our response:

We acknowledge the reviewer's suggestion that gender could have potentially influenced the presentation of our model. However, our analysis did not identify gender as a statistically significant factor, and thus it was not incorporated into the model presented in Table 2.

Comment # 3

I would encourage authors to add univariate analysis in the supplementary tables for transparency of selection of variables in the multivariate analysis.

Our response:

We agree with the reviewer and added the following tables (univariate analysis) as supplementary tables: 

Table 2: Target organ damage

 Infection from pulmonary source

n=44,491 Infection from other sources

n=50,486 P-value

Respiratory failure, No. (%) 2341 (5.3) 522 (1) <0.001

Vascular, No. (%) 303 (0.7) 313 (0.6) 0.24

Renal, No. (%) 3074 (6.9) 4194 (8.3) <0.001

Liver, No. (%) 59 (0.1) 336 (0.7) <0.001

Hematologic, No. (%) 617 (1.4) 631 (1.2) 0.06

Metabolic, No. (%) 358 (0.8) 71 (0.1) <0.001

CNS, No. (%) 24 (0.1) 79 (0.2) <0.001

SIRS, No. (%) 322 (0.7) 252 (0.5) <0.001

Any target organ damage, No. (%) 6230 (14) 5810 (11.5) <0.001

Table 1: Demographic characteristics and hospitalization details 

 Infection from pulmonary source

n=44,491 Infection from other sources

n=50,486 P-value

Age, mean ± SD 70.7 ± 17.3 68 ± 19.4 <0.001

Male sex, No. (%) 23226 (52.2) 23568 (46.7) <0.001

Deceased, No. (%) 26651 (59.9) 26031 (51.6) <0.001

 Age, mean ± SD 81.1 ± 11.9 81.8 ± 11.4 <0.001

 In hosp, No. (%) 1846 (6.9) 890 (3.4) <0.001

 One week disch, No. (%) 478 (1.8) 427 (1.6) 0.18

 One month disch, No. (%) 1782 (6.7) 1779 (6.8) 0.5

ICU transfer, No. (%) 934 (2.1) 547 (1.1) <0.001

Hospitalization length of stay, median (IQR) 5 (3-7) 5 (3-7) <0.001

 

Table 3: Background medical conditions

 Infection from pulmonary source

n=44,491 Infection from other sources

n=50,486 P-value

DM 16147 (36.3) 18821 (37.3) 0.002

IHD 16070 (36.1) 15516 (30.7) <0.001

CVA 2348 (5.3) 2954 (5.9) <0.001

Hematologic malignancy 2350 (5.3) 1812 (3.6) <0.001

Solid malignancy 24835 (55.8) 27775 (55) 0.01

Liver cirrhosis 534 (1.2) 919 (1.8) <0.001

CHF 10899 (24.5) 8434 (16.7) <0.001

CKD/renal failure 10151 (22.8) 11435 (22.6) 0.54

COPD 15304 (34.4) 6200 (12.3) <0.001

HTN 28915 (65) 31893 (63.2) <0.001

Asthma 8111 (18.2) 4649 (9.2) <0.001

A. Fib/Flutter 9596 (21.6) 8114 (16.1) <0.001

Table 4: Chronic medication use

 Infection from pulmonary source

n=44,491 Infection from other sources

n=50,486 P-value

BB 15156 (34.1) 16127 (31.9) <0.001

ACEi 13103 (29.5) 14917 (29.5) 0.75

ARB’s 5246 (11.8) 5161 (10.2) <0.001

Steroids 7367 (16.6) 4648 (9.2) <0.001

Statins 16674 (37.5) 17828 (35.3) <0.001

Altroxin 2775 (6.2) 2951 (5.8) 0.01

Purchased three months before hospitalization

 

Table 5: Lab tests results during hospitalization (limiting values)

 Infection from pulmonary source

n=44,491 Infection from other sources

n=50,486 P-value

Hb (min), gr/dL, mean ± SD 11.4 ± 1.9 11.2 ± 1.9 <0.001

Hb < 6, No. (%) 78 (0.2) 95 (0.2) 0.67

Hb < 7, No. (%) 495 (1.1) 569 (1.1) 0.9

Hb < 9, No. (%) 4888 (11.1) 5997 (12) <0.001

WBC (max), 103/µL, median (IQR) 11.9 (8.8-16.2) 12.6 (9.2-16.8) <0.001

WBC (min), 103/µL, median (IQR) 8.1 (6.1-10.5) 7.9 (6-10.3) <0.001

WBC > 15, No. (%) 13476 (30.6) 17065 (34) <0.001

WBC < 5, No. (%) 6019 (13.9) 6670 (13.4) 0.06

PLT (min), 103/µL, mean ± SD 219.2 ± 87.5 212.5 ± 83.2 <0.001

PLT < 50, No. (%) 742 (1.7) 622 (1.3) <0.001

INR (max), median (IQR) 1.1 (1-1.3) 1.1 (1-1.3) <0.001

Glucose (max), mg/dL, median (IQR) 147 (120-195) 140 (115-186) <0.001

Glucose (min), mg/dL, median (IQR) 97 (84-117) 94 (82-112.7) <0.001

Glucose > 200, No. (%) 10175 (23.3) 10186 (20.6) <0.001

Glucose < 50, No. (%) 545 (1.3) 719 (1.5) 0.008

Glucose < 70, No. (%) 3105 (7.1) 4463 (9) <0.001

Glucose coefficient of variance, median (IQR) 0.2 (0.1-0.3) 0.2 (0.1-0.3) 0.008

AST (max), U/L, median (IQR) 26 (19-41) 25 (18-41) <0.001

ALT (max), U/L, median (IQR) 22 (15-38) 21 (14-36) <0.001

Creatinine (max), mg/dL, median (IQR) 1 (0.8-1.4) 1 (0.8-1.4) <0.001

Maximal CRE/first CRE ratio, median (IQR) 1 (1-1.1) 1 (1-1) <0.001

Maximal CRE/first CRE > 2, No. (%) 515 (1.2) 451 (0.9) <0.001

D-Dimer (max), ng/mL, median (IQR) 656 (5.6-1509.9) 450.5 (1.7-1576) 0.002

Urea (max), mg/dL, median (IQR) 47 (33-72) 44 (30.5-67) <0.001

Albumin (min), gr/dL, median (IQR) 3.3 (2.9-3.7) 3.3 (2.9-3.6) <0.001

Albumin < 1.5, No. (%) 43 (0.1) 46 (0.1) 0.86

Albumin < 2, No. (%) 720 (1.7) 909 (1.9) 0.01

Albumin < 2.5, No. (%) 3437 (8) 4402 (9.2) <0.001

LDH/albumin ratio, median (IQR) 127.5 (101.1-168.5) 119.4 (93.7-158.9) <0.001

LDH/albumin > 150, No. (%) 13241 (34) 12528 (29.3) <0.001

 

Table 6: LDH

 Infection from pulmonary source

n=44,491 Infection from other sources

n=50,486 P-value

 Count, median (IQR) 2 (1-2) 2 (1-3) <0.001

 First (U/L), median (IQR) 418 (344-522) 385 (316-483) <0.001

LDH (first) categories, No. (%) <0.001

< 480 29625 (66.6) 37557 (74.4) 

480 – 700 11425 (25.7) 9910 (19.6) 

700 - 900 2279 (5.1) 1896 (3.8) 

> 900 1162 (2.6) 1123 (2.2) 

7 days after diagnosis

Comment # 4

In the table 2, 3, and 4, authors have added many biomarkers such as Hb, WBC, PLT, and Albumin. And all these biomarkers have significant associations. However, authors have not discussed the potential role/importance of these markers. I would suggest adding more discussion about these biomarkers.

Our response:

We agree with the reviewer and edited the discussion section: 

"…. Although leukopenia is widely recognized in the literature as a marker for mortality in patients with infection(1), our study yielded unexpected results, as leukopenia was not found to be associated with mortality. Instead, it was linked to increased target organ damage. In contrast, anemia and thrombocytopenia, which have been consistently associated with adverse outcomes in previous studies (2), were found to be correlated with both mortality and target organ damage. Similarly, low levels of albumin were significantly associated with all outcomes, as previously demonstrated in a recently published research (3).

1. Samuel H. Belok, Nicholas A. Bosch, Elizabeth S. Klings, Allan J. Walkey. Evaluation of leukopenia during sepsis as a marker of sepsis-defining organ dysfunction. PLoS One. 2021; 16(6): e0252206. doi: 0.1371.0252206.

2. Mehdi Mirsaeidi, Paula Peyrani, Stefano Aliberti, Giovanni Filardo PhD d et al. Thrombocytopenia and Thrombocytosis at Time of Hospitalization Predict Mortality in Patients With Community-Acquired Pneumonia. Chest Original Research Volume 137, Issue 2, February 2010, Pages 416-420

3. Frenkel A, Novack V, Bichovsky Y, Klein M, Dreiher J. Serum Albumin Levels as a Predictor of Mortality in Patients with Sepsis: A Multicenter Study. Isr Med Assoc J, 2022 Jul ;24:454-459

Comment # 5

In the conclusion section, 'Additionally, patients with pulmonary infections and LDH levels >900 U/L are at particularly increased risk of death within 30 days. The findings presented here suggest LDH testing as a helpful bedside tool to estimate mortality risk in patients with infectious disease.' I would suggest rephrasing these sentences, as this study did not assess risk but 'increase odds of death'.

Our response:

 We concur with the reviewer's observation that the original phrasing of the paragraph was imprecise and potentially misleading. Therefore, we have revised the language as follows:

"…. Additionally, the analysis revealed that patients with pulmonary infections and LDH levels greater than 900 U/L had higher mortality odds within a 30-day timeframe. These findings support the utility of LDH testing as a valuable bedside tool for estimating mortality odds in patients with infectious diseases."

Comment # 6

A number of previous studies and reviews have shown that In patients with COVID-19, the increased LDH level is associated with a higher risk of negative clinical prognosis and higher mortality. Current data included the year 2020 in the analysis. Author should add a caveat specific diseases such as COVID-19.

Our response:

We agree with the reviewer, and edited the discussion section accordingly: 

"…It is worth noting that the current study did not involve patients with COVID-19 infection, despite data being collected up until 2020. However, it is noteworthy that recent research has highlighted a significant association between high LDH values and disease severity and mortality among COVID-19 patients. [18]."

[18]. Brandon Michael Henry , Gaurav Aggarwal , Johnny Wong , Stefanie Benoit et al. Lactate dehydrogenase levels predict coronavirus disease 2019 (COVID-19) severity and mortality: A pooled analysis. Am J Emerg Med. 2020 Sep;38(9):1722-1726. doi: 10.1016/j.ajem.2020.05.073. Epub 2020 May 27.

Comment # 7

I will suggest to include more references to support the statements made in the discussion. I will also suggest to review the manuscript for grammar and use same fonts for manuscript and all references.

Our response:

 We appreciate the reviewer's contribution in reviewing the manuscript for grammatical errors and ensuring consistency in fonts. Furthermore, we have incorporated four additional references to support the arguments presented in the discussion, while all grammatical inaccuracies identified during the review process have been rectified.

Samuel H. Belok, Nicholas A. Bosch, Elizabeth S. Klings, Allan J. Walkey. Evaluation of leukopenia during sepsis as a marker of sepsis-defining organ dysfunction. PLoS One. 2021; 16(6): e0252206. doi: 0.1371.0252206.

1) Mehdi Mirsaeidi, Paula Peyrani, Stefano Aliberti, Giovanni Filardo PhD d et al. Thrombocytopenia and Thrombocytosis at Time of Hospitalization Predict Mortality in Patients With Community-Acquired Pneumonia. Chest Original Research Volume 137, Issue 2, February 2010, Pages 416-420

2) Frenkel A, Novack V, Bichovsky Y, Klein M, Dreiher J. Serum Albumin Levels as a Predictor of Mortality in Patients with Sepsis: A Multicenter Study. Isr Med Assoc J, 2022 Jul ;24:454-459

3) Brandon Michael Henry , Gaurav Aggarwal , Johnny Wong , Stefanie Benoit et al. Lactate dehydrogenase levels predict coronavirus disease 2019 (COVID-19) severity and mortality: A pooled analysis. Am J Emerg Med. 2020 Sep;38(9):1722-1726. doi: 10.1016/j.ajem.2020.05.073. Epub 2020 May 27.

Reviewer 3 report:

Thorough, intelligible review of elevated LD in sepsis with meaningful, interesting results.

Possibly limited clinical utility given the increased LD in different kind of sepsis only modest and would add little to clinical evaluation and management.

There is a considerable literature in this area, which they only summarise but usefully compare to other biomarkers.

Our response:

We agree with the reviewer. Following his comment, we expanded the discussion while referring to other biomarkers. We also added four references following this expansion.

1..Samuel H. Belok, Nicholas A. Bosch, Elizabeth S. Klings, Allan J. Walkey. Evaluation of leukopenia during sepsis as a marker of sepsis-defining organ dysfunction. PLoS One. 2021; 16(6): e0252206. doi: 0.1371.0252206.

2. Mehdi Mirsaeidi, Paula Peyrani, Stefano Aliberti, Giovanni Filardo PhD d et al. Thrombocytopenia and Thrombocytosis at Time of Hospitalization Predict Mortality in Patients With Community-Acquired Pneumonia. Chest Original Research Volume 137, Issue 2, February 2010, Pages 416-420

3. Frenkel A, Novack V, Bichovsky Y, Klein M, Dreiher J. Serum Albumin Levels as a Predictor of Mortality in Patients with Sepsis: A Multicenter Study. Isr Med Assoc J, 2022 Jul ;24:454-459

4. Brandon Michael Henry , Gaurav Aggarwal , Johnny Wong , Stefanie Benoit et al. Lactate dehydrogenase levels predict coronavirus disease 2019 (COVID-19) severity and mortality: A pooled analysis. Am J Emerg Med. 2020 Sep;38(9):1722-1726. doi: 10.1016/j.ajem.2020.05.073. Epub 2020 May 27.

---

## [Decision Letter · Decision Letter 1]

8 Mar 2023

­­The role of lactate dehydrogenase in hospitalized patients, comparing those with pulmonary versus non-pulmonary infections: A nationwide study

PONE-D-22-32402R1

Dear Dr. Amit Frenkel,

We’re pleased to inform you that your manuscript has been judged scientifically suitable for publication and will be formally accepted for publication once it meets all outstanding technical requirements.

Kind regards,

Riccardo Nevola, MD, PhD

Academic Editor

PLOS ONE

**Comments to the Author**

1. If the authors have adequately addressed your comments raised in a previous round of review and you feel that this manuscript is now acceptable for publication, you may indicate that here to bypass the “Comments to the Author” section, enter your conflict of interest statement in the “Confidential to Editor” section, and submit your "Accept" recommendation.

Reviewer #2: All comments have been addressed

2. Is the manuscript technically sound, and do the data support the conclusions?

Reviewer #2: Yes

3. Has the statistical analysis been performed appropriately and rigorously? 

Reviewer #2: Yes

4. Have the authors made all data underlying the findings in their manuscript fully available?

Reviewer #2: No

5. Is the manuscript presented in an intelligible fashion and written in standard English?

Reviewer #2: Yes

6. Review Comments to the Author

Reviewer #2: Thanks a lot. Authors have incorporated all the suggested changes in the manuscript. No further comments.

7. PLOS authors have the option to publish the peer review history of their article (what does this mean?). If published, this will include your full peer review and any attached files.

Reviewer #2: **Yes: **Prabal Chourasia

---

## [Editor Report · Acceptance letter]

16 Mar 2023

PONE-D-22-32402R1 

­­*The role of lactate dehydrogenase in hospitalized patients, comparing those with pulmonary versus non-pulmonary infections: A nationwide study*

Dear Dr. Frenkel:

I'm pleased to inform you that your manuscript has been deemed suitable for publication in PLOS ONE. Congratulations! Your manuscript is now with our production department. 

Kind regards, 

on behalf of

Dr. Riccardo Nevola 

Academic Editor

PLOS ONE